# Subcutaneous mycoses: Endemic but neglected among the Neglected Tropical Diseases in Ethiopia

**Wendemagegn Enbiale**[1]*, **Alemayehu Bekele**[2], **Nigus Manaye**[3], **Fikre Seife**[4], **Zeyede Kebede**[3], **Filmon Gebremeskel**[5], **Johan van Griensven**[6]

1 Bahir Dar University, College of Medicine and Health sciences, Bahir Dar, Ethiopia, 2 Arba Minch University Collaborative Research and Training Center for Neglected Tropical Diseases, Arba Minch, Ethiopia, 3 World Health Organization, country office, Addis Ababa, Ethiopia, 4 Ministry of Health, Addis Ababa, Ethiopia, 5 Mekele University, College of Medicine and Health Sciences, Mekele, Ethiopia, 6 Institute of Tropical Medicine, Antwerp, Belgium

* wendemagegnenbiale@gmail.com

## Abstract

### Background

Subcutaneous (deep) mycoses are a chronic infectious disease of the skin and underlying structures endemic in tropical countries. The disease has serious medical and socioeconomic consequences for patients, communities and health services in endemic areas. The inclusion of mycetoma and other subcutaneous mycoses in the list of Neglected Tropical Diseases by WHO highlights the need to assess the burden of these diseases and establish control programs where necessary. In Ethiopia no strategies can be devised because of a lack of epidemiologic information. To address this evidence gap, we performed a national rapid assessment of the geographic distribution of subcutaneous mycoses.

### Methodology

We conducted a rapid retrospective assessment using hospital records to identify all suspected and confirmed cases of subcutaneous mycoses in 13 referral hospitals across the country between 2015 and 2022. In each hospital the logbooks were reviewed for diagnoses of subcutaneous mycosess, as diagnosed per routine practice. Descriptive analysis was done.

### Result

From 13 hospitals we extracted 143 cases of subcutaneous mycoses, registered from July 2018 to September 2022. 118 (82.5%) patients were diagnosed as mycetoma, 21 (14.7%) as chromoblastomycosis and the remaining 4 (2.8%) as sporotrichosis. The mean age of patients was 35.8 years (SD = 14.5). 101 (70.6%) patients were male and 96 (67.1%) patients were farmers. 64 (44.8%) cases were from the Tigray regional state. 56 (65.9%) patients had information on diagnostic microscopic evaluation: for mycetoma histopathologic evaluation and fine needle aspiration cytology had a higher positivity rate while for

**Data Availability Statement:** All relevant data are within the paper and its Supporting Information files.

**Funding:** The author(s) received no specific funding for this work.

**Competing interests:** The authors have declared that no competing interests exist.

chromoblastomycosis potassium hydroxide (KOH) staining had a better yield. The main clinical presentations were nodules, sinuses and infiltrative plaques on the skin. Radiologic findings of bone involvement was present in some.

## Conclusions

Mycetoma and other subcutaneous mycoses are endemic in Ethiopia, with cases reported from almost all regions with the highest cases numbers reported from the northern part of the country. A routine program and systems should be developed to identify and document the burden of subcutaneous fungal infections in the country. Diagnosis and treatment guidelines should be developed.

### Author summary

Subcutaneous mycoses (fungal infections) are a chronic infectious disease of the skin and underlying structures endemic in tropical countries. These diseases are considered neglected tropical diseases, with significant health and socioeconomic impacts on patients and communities. However, there is a lack of epidemiological information about these diseases in Ethiopia. To address this gap, a national rapid assessment was conducted using hospital records to identify all suspected and confirmed cases of subcutaneous mycoses in 13 hospitals in different regions of the country between 2015 and 2022.

The rapid assessment conducted in Ethiopia revealed that the country is endemic for mycetoma and other subcutaneous mycoses, with cases reported from almost all regions but with variations in case distribution. This study constitutes the largest number of cases reported from Ethiopia, and the first to cover all main geographical areas across the country. More than two third of cases reported are from the northern part of the country (Tigray and Amhara region) alone, despite the fact that these regions comprise only about a quarter of the total population of the country. At the same time, the smaller number of cases (143) registered in almost 5 years indicate a lower prevalence compared to Sudan, a neighbouring country which has reported more than 9600 cases over a span of 30 years.

The authors noticed, there is a lack of research and surveillance programs for subcutaneous mycoses, and more accurate data and national mapping of the disease are needed. A routine program and systems should be developed to identify and record cases of subcutaneous mycoses at all levels of healthcare, including primary healthcare, and public awareness and education efforts should be intensified.

In conclusion the findings of this rapid assessment underscore the need for increased attention to subcutaneous mycoses in Ethiopia. The scarcity of research and surveillance programs for these infections highlights the urgent need for developing routine programs and systems for identifying and recording cases of subcutaneous mycoses in health care system, along with revising the health information system indicators to capture skin neglected tropical diseases, would be valuable steps. These efforts should be combined with public awareness and education campaigns targeting community workers and health professionals to improve identification and referral of patients.

## Introduction

*Subcutaneous (deep) mycoses* are a chronic infectious disease of the skin and underlying structures caused by a diverse group of microorganisems that produce disease when traumatically introduced into the skin and subcutaneous tissue [1]. Mycetoma, chromoblastomycosis and sporotrichosis are the most common subcutaneous mycoses. Subcutaneous mycoses are a neglected tropical disease (NTD) that, like other NTDs, have numerous health and socioeconomic impacts on patients and communities [2,3]. While subcutaneous mycoses can affect individuals of all ages, those in reproductive age and the economically most active age groups are most commonly affected [3]. A study on the consequences of mycetoma showed that about 83.7% of mycetoma patients reported disability resulting in loss of function and for some amputation. The disease prevents patients from attainment of education, causes a significant financial burden to affected populations and interferes with employment [4].

Subcutaneous mycoses are endemic in tropical and subtropical regions [2], primarily favorable humidity and temperature conditions that promote the growth and spread of the causative agent. Several of these agents have specific geographic distributions with certain variation in clinical presentation and management approches [3].

Mycetoma is caused by diverse microorganisms, including both bacteria (actinomycetoma) and fungi (eumycetoma) [1,2,3,5]. The causative agents are found in hot and humid climate zones (tropics) in the soil, plants, woods, decaying plant material, and compost. This explains the higher occurrence of the disease in farmers and agricultural workers. Classical cases present with large tumor-like swellings, characterized by a triad of tumefaction, draining sinuses, and the presence of grains [3]. The lower extremities are most commenly afected by the disease [2,3,5,6].

Chromoblastomycosis is caused by various pigmented or dematiaceous fungi [7,8]. The disease predominantly occurs in tropical and subtropical regions but has a worldwide distribution [9,10]. Infection usually occurs in adulthood between the age of 20 and 40 years with a higher prevalence among men, which may reflect occupational activities as a risk factor [6,9,11]. Clinically it presents initially with unilateral and asymmetric scaling cutaneous-subcutaneous nodules, evolving into verrucous lesions, vegetating plaques, scaling cauliflower-like tumors; crusts and ulcers can develop.

Sporothrichosis is caused by dimorphous fungi of the *Sporothrix schenckii* complex and primarily affects the skin and lymphatic vessels. The disease manifests in the form of subcutaneous nodules or gummata, with rare involvement of bone, joints or internal organs [8,11]. While it has a global distribution, most cases are reported from South American countries. The pathogens of the *Sporothrix schenckii* complex develop in plants, wood, decaying plant materials in warmer and humid climates [8]. Inoculation usually occurs through injury. Children and young adults are commonly affected with almost no gender difference [8,11].

The etiological diagnosis of subcutaneous mycoses poses challenges. Culture of grains, or more recently, molecular methods, are considered the best diagnostic approches, as they allow for species identification. However, these methods have limited availability in resource-constrained settings. In such settings, more accessible laboratory and imaging investigations can provide support for the clinical diagnosis. These include microscopic evaluation of material drained from the sinuses, fine needle aspiration (FNA) or biopsy. Potassium hydroxide (KOH) staining is often performed, to digest keratin and provide a clearer background to visualizing fungal elements. Gram-staining can also be useful, as detection of filamentous bacteria is highly suggestive of actinomycetoma. Whereas X-ray and ultrasound can help determine the extent of lesions, ultrasound can also assist in the diagnosis of mycetoma [1,5,6].

The inclusion of mycetoma in the list of NTDs at the 69th World Health Assembly emphasizes the need to assess the burden of these diseases and establish control programs where

necessary [12]. Despite this recognition by the World Health Organization (WHO) in 2016 and the subsequent global initiatives to estimate the global burden and map the distribution of the disease, little progress has been made in addressing the evidence gap over the past six years. Consequently, there is still a lack of basic epidemiologic data. One of the reasons for this is that mycetoma and other skin subcutaneous mycoses are not notifiable diseases, resulting in the absence of surveillance and routine data registration systems [13]. Sudan is the only country to have a national mycetoma control program [14,15], while many other countries, including Ethiopia, lack strategies due to lack of epidemiologic information.

The Third National Neglected Tropical Diseases Strategic Plan of Ethiopia (2021–2025) includes mycetoma as one of the NTDs without established endemicity in the country. With the available evidence of only 20 cases reported from a single institution in Ethiopia, the strategic plan emphasizes the necessity of disease mapping [16,17]. To address this evidence gap, we conducted a national rapid assessment to determine the geographic distribution of subcutaneous mycoses in Ethiopia.

## Methodology

### Ethics statement

The assessment was commissioned by the WHO regional office in collaboration with the Ethiopian Federal Ministry of Health (MOH). We obtained support letter from the MOH for the respective hospitals, allowing accessing the Health Information Management System and outpatient clinical records. Any information that could identify patients was removed from the data analysis and result to ensure privacy and confidentiality were taken. As the project was an integral part of the MOH for health system planning a formal ethical clearance was not needed.

### Study design and selection of hospital

We conducted a cross-sectional study using hospital records to identify all suspected and confirmed cases of subcutaneous mycoses in 13 referral hospitals across the country recorded between 2015 and 2022.

### General setting

Ethiopia is the second-most populous country in Africa. The country has a population of about 117,000,000 with 20.9% of the population living in urban areas. The country is geographically and politically administered in 11 regions (of which six with a large surface) and two special administration cities. The country is one of the poorest, with a per capita income of $960 and is amongst the lowest ranked ones in Africa for total health expenditure [18]. The country has one of the lowest health workforce densities (1.4 per 1000) in sub-Saharan Africa. [17,19]. Ethiopia has a three-tier health care system, with primary, secondary and tertiary levels of care. The primary level includes primary hospitals, health centers and health posts, the secondary level entails district hospitals and the tertiary level concerns referral and specialty hospitals [20].

### Selection of hospitals, identification of cases, data collection and analysis

For the retrospective data collection, we included the referral hospitals or dermatology centers in the regions or administrative towns, as they are the main referral diagnostic and treatment facilities in the regions and hence better placed to diagnose rare or difficult to diagnose conditions, and with better documentation. A total of 13 hospitals were selected in the seven regions and the two administrative cities (Table 1).

**Table 1. Hospitals in Ethiopia visited from September 20 to November 2, 2022 for the national subcutaneous mycoses survey.**

| | Region | Town | Hospital | Catchment population |
|---|---|---|---|---|
| 1 | Amhara | Bahir Dar | Tibebe Ghion specialized hospital* | 3 to 5 million |
| | | Boru Meda | Boru Meda primary hospital* | 1.5 million |
| 2 | Afar | Dubuti | Dubti referral hospital | 1.5 million |
| | | Amibara | Mohammed Akile memorial general hospital | 1.5 million |
| 3 | Somali | jigjga | Karama general hospital | 1.5 million |
| | | | Jigjga University referral hospital* | 3 to 5 million |
| 4 | Southern nation nationality people (SNNP) | Arba Minch | Arba Minch general hospital* | 1.5 million |
| | | Sodo | Wolayita otona referral hospital* | 3 to 5 million |
| 5 | Harari | Harar | Hiwot Fana hospital* | |
| 6 | Oromia | Bisidemo | Bisidemo general hospital* | 1.5 million |
| 7 | Tigray | Mekele | Ayder specialized hospital* | 3 to 5 million |
| 8 | Addis Ababa | Addis Abeba | All-African Leprosy Rehabilitation and Training Centre (ALERT) hospital* | 3 to 5 million |
| 9 | Drie Dawa | Drie Dawa | Dilechora general hospital* | 1.5 million |

• *Hospitals with dermatologist and dermatology services

In each hospital or center, we first queried physicians providing dermatological services about the entry point and registration of potential cases of subcutaneous mycoses. Consistently, this entailed the outpatient department, with the logbook as the main source of data. This book contains socio-demographic information combined with the final diagnosis. Between September 20 and November 2, 2022, all available logbooks from 2015 on were reviewed for diagnoses of subcutaneous mycoses, as diagnosed per routine practice. The diagnosis of subcutaneous mycosis was considered if the register mentioned: mycetoma, chromoblastomycosis, sporotrichosis and subcutaneous mycoses as diagnosis. The laboratory and imaging information's were collected (if retrievable) as supportive evidences. For each case, the patient files were retrieved and reviewed. Data extracted included sociodemographic data, duration and site of the lesion(s), history of trauma, diagnostic methods used and results, and other associated pathologies at the time of presentation to the hospital. For the laboratory investigations, we recorded whether a gram-stain or KOH staining was done on aspirated material. A positive gram-stain test implies that filamentous bacteria were seen, suggestive of actinomycetoma; a positive KOH test implies that fungal elements were seen, suggestive of eumycetoma. For FNAC, gram stain and Periodic Acid Shiff (PAS) was done; for biopsies, hematoxylin/eosin and PAS staining was usually done.

Data were collected on paper, and then entered into an excel database for analysis. Analysis was purely descriptive. Binary/categorical data were presented as frequencies and percentages; continuous variables were summarized using means and standard deviation.

## Results

### Sociodemographic characteristics

From 13 hospitals across 7 regions and two administration cities, we extracted 143 cases of subcutaneous mycoses, registered from July 2018 to September 2022. The mean age of patients was 35.7 years (SD = 14.5), the youngest case was 10 years old and the oldest 70 years. 101 (70.6%) patients were male. 96 (67.1%) were farmers and 27 (18.9%) students. 64 (44.8%) cases were from the Tigray region in the north of the country regional state and the region with the lowest case numbers was Benishangule regional, in the west of the country (Fig 1).

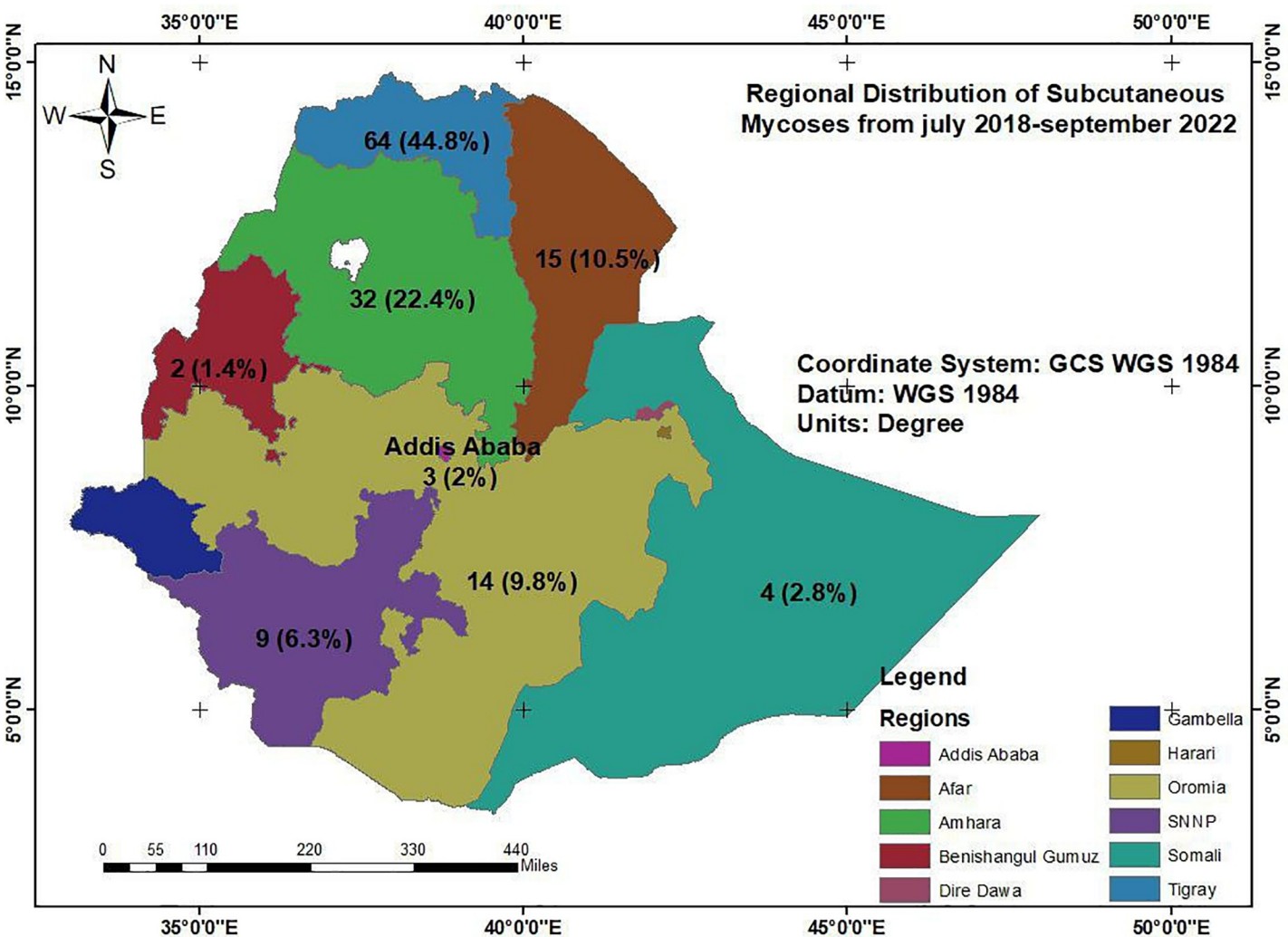

**Fig 1. Regional distribution of subcutaneous mycoses in Ethiopia, seen at referral hospitals from data registered from July 2018 to September 2022 (Source Ethio_GIS, 2019).**

**Clinical presentation.** The mean time of presentation to the health facility was 5 years (SD 4.4 years). For 58 patients with information on whether there was a history of trauma at the site of the lesion, 33 (56.9%) reported a trauma preceding the lesion. 118 (82.6%) patients with subcutaneous mycoses were diagnosed as mycetoma (Fig 2), 21 (14.6%) as chromoblastomycosis and the remaining four (2.8%) as sporotrichosis (Fig 3).

56 (65.9%) patients had recorded information on diagnostic microscopic evaluation: 34 patients had a KOH examination done, 17 a gram stain, 27 biopsy and 18 FNAC. In 49 (57.7%) cases, the diagnosis was supported by the microscopic findings. About half of the histopathological examinations reported isolation of grains in mycetoma cases. From the 16 mycetoma cases with further differentiation, ten were labeled as eumycetoma and six as actinomycetoma.

For mycetoma, histopathologic evaluation and FNAC had a higher positivity rate while for chromoblastomycosis KOH had a higher yield (Table 2). One third (n = 29) of the patients had information on the x-ray findings and 9 (31.0%) of them had lytic bone involvement or apparent osteomyelitis.

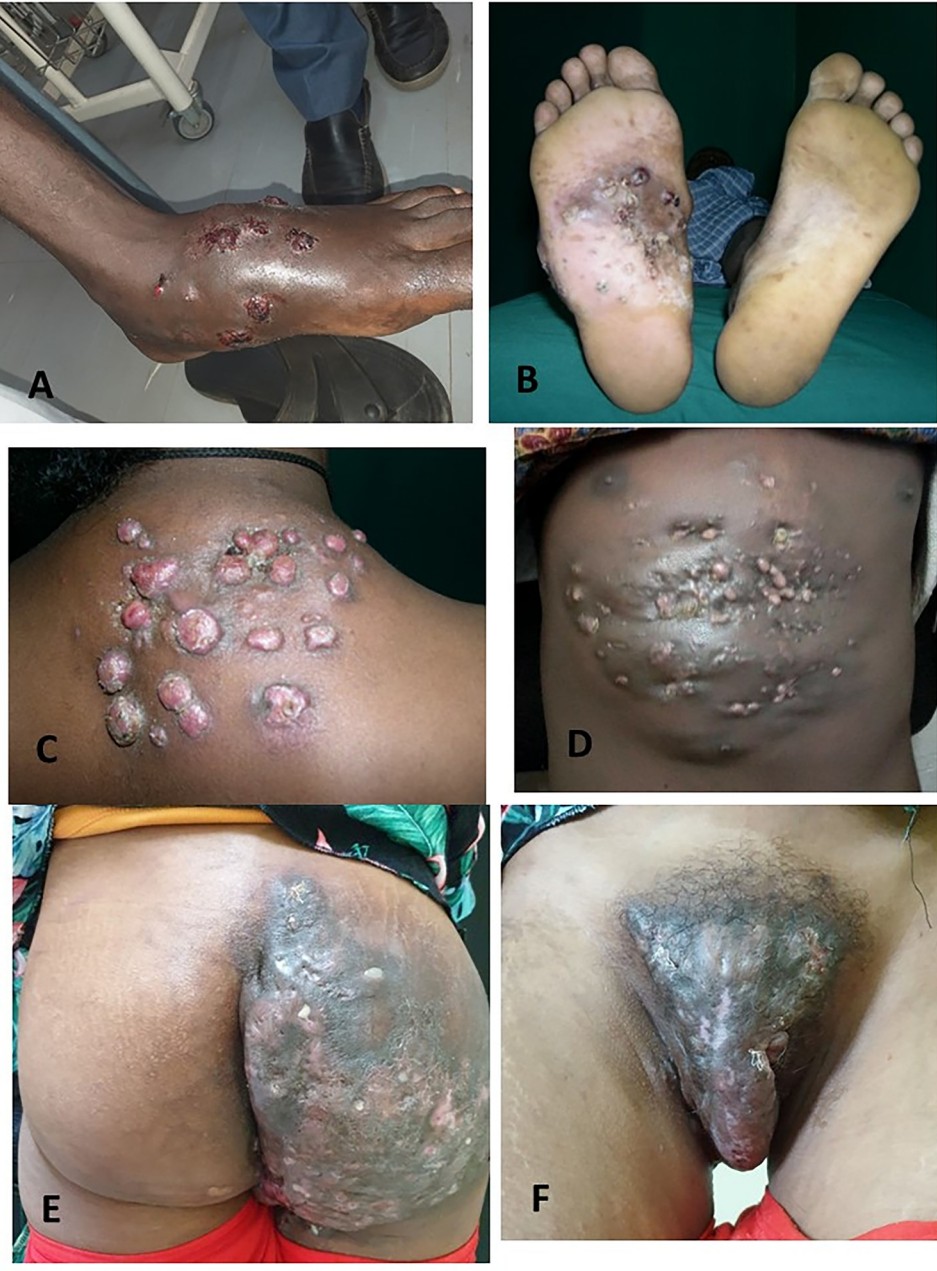

**Fig 2. Mycetoma and the different body site affected from cases seen in referral hospitals in Ethiopia.** A) Woody hard globular swelling of the left foot with multiple sinuses. B) Infiltrative plaques on the sole of the left foot medially with sinuses and grains. C) Multiple nodules and sinuses on the right upper back extending to the shoulder. D) Hard to rubbery infiltrative plaque on the lower chest and upper abdomen with papules, nodules and sinuses. E) & F) A patient with both buttock and genital involvement. Infiltrative mass on the right buttock with multiple sinuses and hyperpigmented infiltrative involvement of the pubic area of the external genitalia with tumorous enlargement of the labia.

In about 85% of the cases with documented information, the primary site of the subcutaneous mycosis was at the foot or leg, but cases involving the arm, buttock, genital area and trunk were also registered (Fig 2). The main clinical presentations ware nodules, sinuses and infiltrative plaques on the skin.

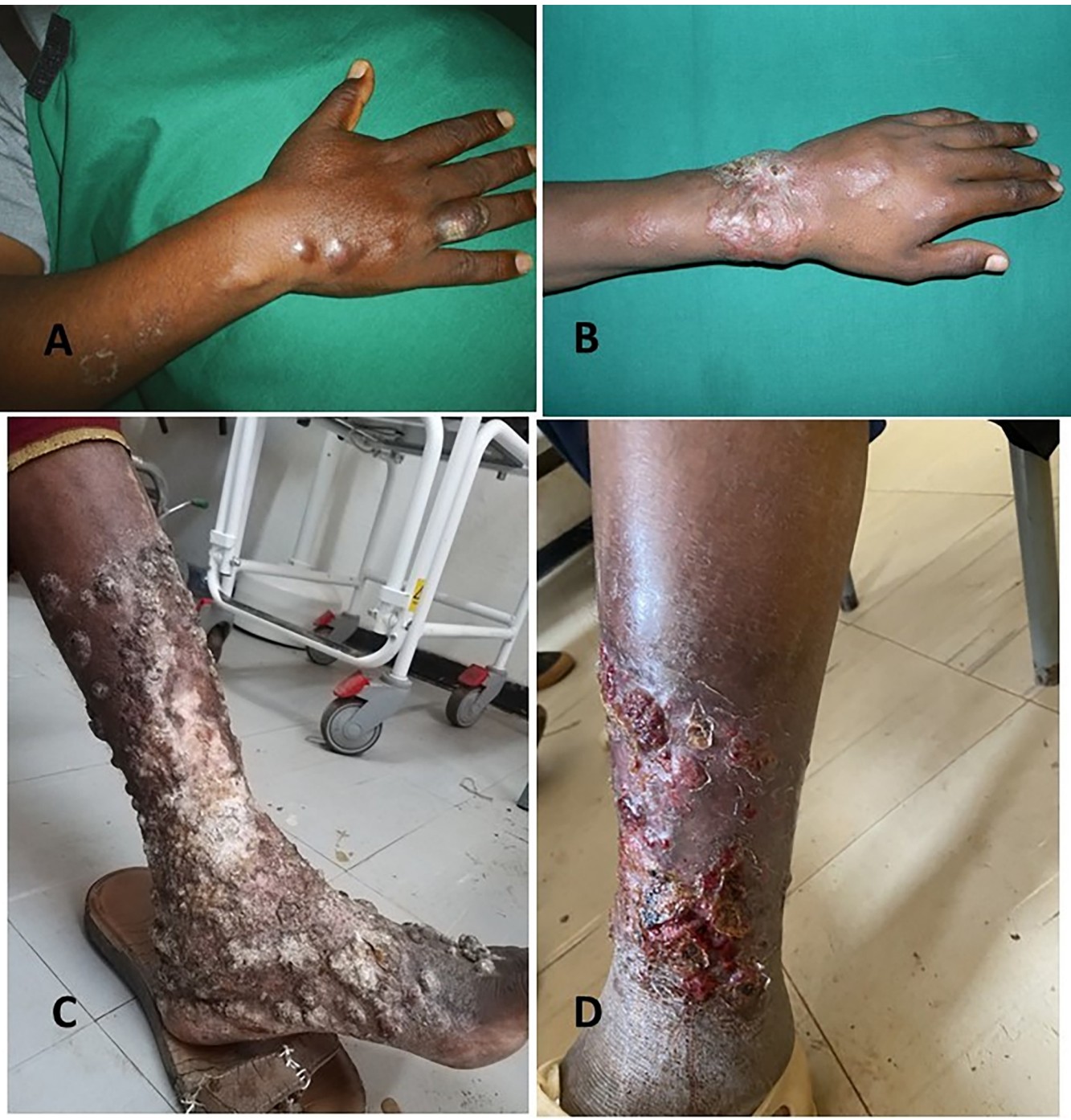

**Fig 3.** A) Sporotrichosis; swollen ring finger with crusting, nodular lesions on dorsum of hand and wrist linearly arranged. B) Sporotrichosis; infiltrative plaque with satellite papules. C) Chromoblastomycosis; warty papules and nodules covering the lower leg and dorsum of foot. D) Chromoblastomycosis; Ulcer, hemorrhagic crust and warty papules on the lower leg.

## Discussion

This rapid assessment shows that subcutaneous fungal infections are endemic in Ethiopia and cases are reported from all parts of the country. We observed some regional variation, with

**Table 2. Documented microscopic laboratory investigations done and positivity rate for subcutaneous mycoses from data registered from July 2018 to September 2022.**

| Diagnosis (total) | Diagnostic procedure | | | | | |
|---|---|---|---|---|---|---|
| | KOH on aspirate | | Gram stain on aspirate | | Histopathology/FNAC | |
| | Number done | % positive | Number done | % positive | Number done | % positive |
| Mycetoma (118) | 26 | 50.0 | 15 | 40 | 27 | 76 |
| Chromoblastomycosis (17) | 8 | 100 | 2 | 0 | 7 | 86 |
| Sporotrichosis (4) | 0 | 0 | 0 | 0 | 4 | 75 |

FNAC: Fine Needle Aspiration Cytology; KOH: Potassium Hydroxide

more than two thirds of cases reported from the Tigray and Amhara regions in the north of the country. At the same time, the smaller number of cases (143) registered in almost 5 years indicate a lower prevalence compared to Sudan, a neighbouring country which has reported more than 9600 cases over 30 years [21].

This study constitutes the largest number of cases reported from Ethiopia, and the first to cover all main geographical areas across the country. A case series on subcutaneous mycoses from one of the dermatology referral hospitals in the country (ALERT) located in the capital reported 20 cases documented in five years and most patients (around three-fifths) were diagnosed with chromoblastomycosis [16]. Despite Ethiopia being located in the mycetoma belt, data to support the endemicity was lacking. The ALERT study and our assessment provides the first data on the existence of mycetoma in the country. In our study, we documented three types of subcutaneous mycoses (mycetoma, chromoblastomycosis and sporotrichosis), and most patients (around four-fifths) were diagnosed as mycetoma. The relatively higher proportion of mycetoma cases in our study compared to the ALERT study is probably due to the fact that the latter study included only cases with a confirmed diagnosis (based on microscopy, FNAC and biopsy). As laboratory confirmation of mycetoma is more difficult compared to other subcutaneous mycoses, some mycetoma cases may have been excluded in the ALERT study [22]. Another point to consider is that in ALERT study most (around three quarters) chromoblastomycosis cases came from only one region (Southwest of Ethiopia) [16]. In contrast, the geographic distribution in our study might better represent the national distribution.

The male predominance with close to 90% of the cases aged between 18 year and 60 years old, and with predominant involvement of the feet is likely due to an increased risk of occupational exposure while working in the fields, as also observed in other studies [23–25]. The regions with the highest number of subcutaneous mycoses cases in Ethiopia are those where podoconiosis is endemic. Both diseases have walking barefoot as a shared risk factor, highlighting the need for intensified community awareness raising about wearing protective shoes as primary prevention measure for both diseases [26,27].

During the rapid assessment, we identified important challenges, precluding an accurate estimate of the burden of subcutaneous mycoses. First, records of subcutaneous mycoses could only be extracted from the log books of the different hospitals, staffed by dermatologists or general practitioners trained in dermatology. The Health Management Information System (HMIS) or the hospital electronic medical record system of all the hospitals visited does not contain any record or indicator for most of the skin diseases including subcutaneous mycoses.

Second, in two thirds of the patients where records could be found, the diagnosis was clinical, or at least no laboratory test results could be retrieved. Microscopy, aspiration results, histology and X-ray results were found for around half of the patients, and there was no information on species identification. All the cases from Tigray lacked laboratory information,

as this was difficult to organize due to the conflict. The ALERT study included only confirmed cases, but similarly as in our assessment no culture or molecular study for species identification was done [16]. Another publication from a dermatologic center from Northern Ethiopia reported seven cases of presumed mycetoma but all of these were diagnosed on clinical grounds [28]. One of the programmatic challenges in subcutaneous mycoses is that there is no highly accurate point of care diagnostic method. The existing diagnostic tools to demonstrate the organisms by culture or molecular techniques (PCR) are available only in the reference laboratory in the capital and this service is not usually accessible for routine care.

Radiologic examination was reported for one third of cases, but there was no single report on the use of ultrasound and other imaging techniques. Experts from the mycetoma research center in Sudan have shown that mycetoma has distinctive ultrasonographic features and can differentiate between fungal and bacterial causes and between mycetoma and other non-mycetoma lesions [29, 30]. The technique is simple, non-invasive, quick, reproducible, acceptable to patients and also a more accessible tool for endemic regions, which would be worth adopting in Ethiopia as well.

For all these reasons, our hospital based data can only serve as a proxy indicator. While the data are surely subject to bias and not reflecting the true population burden, they can nevertheless help to put Ethiopia on the mycetoma map. Taking in to consideration that the country is located in the mycetoma belt and shares a border of around about 750km with Sudan—a country which has reported the largest case in the world [15], the true burden in Ethiopia can be expected to be much higher. Currently the country does not have a program for prevention or treatment of subcutaneous mycoses but there is an initiative at the Ministry level to include mycetoma and other subcutaneous mycoses as priority in the next national NTD strategic plan [17]. The results of this rapid assessment can be used as baseline information for widescale national disease mapping, with a focus in the identified high burden areas.

There are a number of important limitations to acknowledge. First, we focussed on referral hospitals or specialist dermatology clinics as this is the place best equipped to make a correct diagnosis, combined with increased awareness and knowledge about subcutaneous mycoses amongst treating physicians. For a first ever national rapid assessment aiming to provide the first evidence of the presence of these conditions across the country, this seems a fair approach. However, the cases reported can only be considered as the tip of the iceberg, as many patients with subcutaneous mycoses might never reach these facilities, or some cases might have been missed or not reported in the logbooks. Second, even at the referral hospital level, most of the diagnoses in this rapid assessment were diagnosed clinically, and hence highly dependent on the skills of the clinician. While several laboratory tests used provided strong support for the diagnosis, these were not always done or either done but perhaps not recorded. Access to fungal culture or molecular tests would have allowed to strengthen the diagnosis and for instance for mycetoma, would have allowed species identification and further characterization of the mycetoma cases as eumycetoma or actinomycetoma. Finally, existing recodes in this retrospective assessment had only limited information on the clinical presentation, treatment provided and treatment response.

## Conclusion

This rapid assessment establishes Ethiopia as endemic region for mycetoma and other subcutaneous mycoses, with varying case distribution across the country. This rapid surveillance has identified a striking scarcity of research and surveillance programmes for subcutaneous fungal infections. To address these issues, accurate and comprehensive data collection, including national disease mapping, should be prioritized. Developing routine programs and systems to identify and record cases of subcutaneous mycosis at all levels of healthcare, along with

revising indicators in the HMIS/DHIS, is crucial to capture skin NTDs in general. This should be combined with intensified efforts in terms of public awareness and education of community workers and health professionals at existing health centers on the identification of subcutaneous fungal infections and subsequent referral of patients.

A prospective study to investigate the risk factors, factors associated with delayed presentation of patients, the diagnosis, treatment and the various medical health and social consequences of subcutaneous fungal infections would be valuable as well. Most importantly, a diagnosis and treatment guideline needs to be developed. In an attempt to address these problems, the ministry needs to use the opportunity of the existing primary health care system and adopting an integrated skin NTDs management model at the primary health care level.

## Supporting information

**S1 Dataset. Data set of the national subcutaneous mycoses survey in Ethiopia, last updated June, 2023.**
(XLSX)

## Author Contributions

**Conceptualization:** Wendemagegn Enbiale.

**Data curation:** Wendemagegn Enbiale.

**Formal analysis:** Wendemagegn Enbiale.

**Funding acquisition:** Wendemagegn Enbiale.

**Investigation:** Wendemagegn Enbiale, Alemayehu Bekele.

**Methodology:** Wendemagegn Enbiale, Johan van Griensven.

**Project administration:** Wendemagegn Enbiale, Alemayehu Bekele, Filmon Gebremeskel.

**Resources:** Wendemagegn Enbiale.

**Supervision:** Wendemagegn Enbiale, Zeyede Kebede.

**Validation:** Wendemagegn Enbiale, Nigus Manaye, Fikre Seife, Zeyede Kebede, Filmon Gebremeskel, Johan van Griensven.

**Visualization:** Wendemagegn Enbiale, Nigus Manaye, Fikre Seife, Zeyede Kebede, Johan van Griensven.

**Writing – original draft:** Wendemagegn Enbiale.

**Writing – review & editing:** Wendemagegn Enbiale, Alemayehu Bekele, Nigus Manaye, Fikre Seife, Zeyede Kebede, Filmon Gebremeskel, Johan van Griensven.

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
