## [Decision Letter · Decision Letter 0]

1 Jun 2023

Dear Dr Enbiale,

Thank you very much for submitting your manuscript "Subcutaneous mycoses: endemic but neglected among the Neglected Tropical Diseases in Ethiopia" for consideration at PLOS Neglected Tropical Diseases. As with all papers reviewed by the journal, your manuscript was reviewed by members of the editorial board and by several independent reviewers. The reviewers appreciated the attention to an important topic. Based on the reviews, we are likely to accept this manuscript for publication, providing that you modify the manuscript according to the review recommendations. 

Sincerely,

Joshua Nosanchuk, MD

Section Editor

Reviewer's Responses to Questions

**Key Review Criteria Required for Acceptance?**

**Methods**

-Are the objectives of the study clearly articulated with a clear testable hypothesis stated?

-Is the study design appropriate to address the stated objectives?

-Is the population clearly described and appropriate for the hypothesis being tested?

-Is the sample size sufficient to ensure adequate power to address the hypothesis being tested?

-Were correct statistical analysis used to support conclusions?

-Are there concerns about ethical or regulatory requirements being met?

Reviewer #1: The objectives of the study are clearly articulated. The study design, retrospective review of hospital records, is appropriate for the stated objectives. The target population is clearly described from the hospital record. This is simply a review of hospital records and no sample size determination was applied. No ethical concerns.

Reviewer #2: (No Response)

**Results**

-Does the analysis presented match the analysis plan?

-Are the results clearly and completely presented?

-Are the figures (Tables, Images) of sufficient quality for clarity?

Reviewer #1: The results of the hospital record reviews were properly presented and analyzed. The available data from the surveyed hospitals was clearly presented. There are figures and tables included in the manuscript.

Reviewer #2: (No Response)

**Conclusions**

-Are the conclusions supported by the data presented?

-Are the limitations of analysis clearly described?

-Do the authors discuss how these data can be helpful to advance our understanding of the topic under study?

-Is public health relevance addressed?

Reviewer #1: Conclusions were supported by the data presented. Limitations of the study were also described. The public health relevance of the rapid assessment to determine the endemicity of mycetoma has been clearly stated.

Reviewer #2: (No Response)

**Editorial and Data Presentation Modifications?**

Reviewer #1: There are some typos and grammatical errors that need to be corrected. There were also some numbering errors in the list of references.

Reviewer #2: (No Response)

**Summary and General Comments**

Reviewer #1: This rapid assessment is very helpful for the national NTD program as a starting point in determining the endemicity of Mycetoma in the country. It can also be a good resource to guide the national prevalence mapping of the disease and to plan appropriate diagnostic and therapeutic interventions for the treatment and control of the disease.

Reviewer #2: (No Response)

PLOS authors have the option to publish the peer review history of their article (what does this mean?). If published, this will include your full peer review and any attached files.

Reviewer #1: Yes: Dr. Teshome Gebre

Reviewer #2: No

Figure Files:

Data Requirements:

Reproducibility:

References

---

## [Editor Report · Decision Letter 1]

21 Aug 2023

Dear Dr Enbiale,

Thank you for the thoughtful and rigorous revision of your manuscript 'Subcutaneous mycoses: endemic but neglected among the Neglected Tropical Diseases in Ethiopia.' We are pleased to inform you that the manuscript has been provisionally accepted for publication in PLOS Neglected Tropical Diseases. I personally enjoyed reading this work!

Best regards,

Joshua Nosanchuk, MD

Section Editor

---

## [Editor Report · Acceptance letter]

19 Sep 2023

Dear Dr Enbiale,

We are delighted to inform you that your manuscript, "Subcutaneous mycoses: endemic but neglected among the Neglected Tropical Diseases in Ethiopia," has been formally accepted for publication in PLOS Neglected Tropical Diseases.

Best regards,

Shaden Kamhawi

co-Editor-in-Chief

Paul Brindley

co-Editor-in-Chief
